

# Demography and predatory potential of *Orius strigicollis* on eggs of *Plutella xylostella* at two temperatures

Shakeel Ur Rehman[1,2], Xingfu Jiang[2], Mahnoor Saleem[3], Xingmiao Zhou[1], Bangqing Chen[4], Khalid Ali Khan[5,6], Ibrahim Osman Ibrahim[2] and Hamed A. Ghramh[5,7]

[1] Hubei Insect Resources Utilization and Sustainable Pest Management Key Laboratory, College of Plant Science and Technology, Huazhong Agricultural University, Wuhan, Hubei, China
[2] State Key Laboratory for Biology of Plant Diseases and Insect Pests, Institute of Plant Protection, Chinese Academy of Agricultural Sciences, Beijing, China
[3] Institute of Computer Science and Technology, Women University, Multan, Pakistan
[4] Dalaoling Nature Reserve Administration of Yichang Three Gorges, Yichang, China
[5] Center of Bee Research and its Products, Unit of Bee Research and Honey Production, Research Center for Advanced Materials Science (RCAMS), King Khalid University, Abha, Saudi Arabia
[6] Applied College, King Khalid University, Abha, Saudi Arabia
[7] Biology Department, Faculty of Science, King Khalid University, Abha, Saudi Arabia

Corresponding authors
Xingmiao Zhou,
xmzhou@mail.hzau.edu.cn
Bangqing Chen, 380464642@qq.com

## ABSTRACT

**Background.** The polyphagous predatory bug *Orius strigicollis* Poppius (Heteroptera: Anthocoridae) is an active predator used to control many insect pests of agricultural crops. *Orius* species are significantly affected by the type of food and temperature.
**Method.** A study of *O. strigicollis* feeding on *Plutella xylostella* L. (Lepidoptera: Plutellidae) eggs in climate chambers at 28 and 32 °C, 70 ± 5% relative humidity, 16:8 photoperiod, was conducted to determine the effects of different temperatures on the predation activity, biological characteristics and demographic parameters of *O. strigicollis*. Twosex-MS Charts were used to determine the age-stages and characteristics of this species.
**Results.** The results showed that the daily consumption of pre-adults on eggs of *P. xylostella* was highest at 28 °C, and at this temperature, there was a greater probability that *O. strigicollis* would survive to adulthood (42.5%) than at 32 °C (25.0%). It has also been found that at 28 °C there was a long oviposition period (9.38 days) and the greatest female fecundity (44.2 eggs/female) In addition to the highest life expectancy of *O. strigicollis* (16.96 days) at 28 °C, the intrinsic rate of increase (0.087 d$^{-1}$) was also highest. According to our results, *O. strigicollis* has the potential to grow and develop on the eggs of *P. xylostella* at 28 °C and, therefore, could potentially be used as a biological control agent in integrated pest management (IPM) programs.

## INTRODUCTION

Diamondback moth (*Plutella xylostella* L., Lepidoptera: Plutellidae) is a cosmopolitan insect pest that mainly feed on cruciferous crops like broccoli, cabbage, turnips, and cauliflower. It is widely recognized as one of the most common insect pests globally (*Garrad, Booth & Furlong, 2016*; *Shakeel et al., 2017*; *Steinbach, Moritz & Nauen, 2017*). The larval stage is polyphagous, has a short generation time, higher fecundity, exceptional ability to survive under intensive temperatures and insecticide resistance (*Furlong, Wright & Dosdall, 2013*; *Gu et al., 2010*; *Shelton & Nault, 2004*; *Zalucki et al., 2012*). Newly hatched larvae mine through the spongy mesophyll tissues and initiate damage (*Harcourt, 1957*). However, significant damage is caused by the second to fourth larval stages. These stages feed directly from flowers, leaves, buds, the green outer layer of stems, siliques, and on developing seeds within older siliques (*Sarfraz, Keddie & Dosdall, 2005*). *P. xylostella* is a multivoltine pest, and in tropical and temperate regions, it could have more than 20 generations per year (*Harcourt, 1986*). An individual female can lay over 200 eggs on the upper leaf surface of plants (*Justus, Dosdall & Mitchell, 2000*; *Talekar et al., 1994*). The outbreaks of *P. xylostella* are widely distributed in China, and they are often associated with severe losses in the production of cruciferous crops (*Feng et al., 2011*). There is also evidence that the populations found in the southern parts of China migrated northwards and have become a serious pest of many vegetable crops to the north parts of the country (*Yang et al., 2015*). The estimated damage caused by *P. xylostella* to *Brassica* vegetable crops in China gradually increased from 0.15 million hectares to 2.23 million hectares in the last few years (*Li et al., 2016*). It has been reported that the use of insecticides systematically and vigorously to control populations of *P. xylostella* in tropical and subtropical areas has led to a higher selection of insecticide resistant (*Salman, Aydınlı& Ay, 2015*; *Shaaya et al., 1997*; *Talekar & Shelton, 1993*). Therefore, it can be argued that biological control with the use of natural enemies and botanical extracts can serve as a beneficial alternative to synthetic pesticides as part of integrated pest management programs against *P. xylostella* outbreak (*Symondson, Sunderland & Greenstone, 2002*).

In the family of Anthocoridae, *Orius* is considered as one of the largest genus of flower bugs. There are over 80 species in the world that feed on a wide range of small insects that are pests in forests and agricultural crops (*Carpintero, 2002*; *Hernández, 1999*; *Herring, 1966*; *Postle, Steiner & Goodwin, 2001*; *Yamada, Yasunaga & Artchawakom, 2016*). *Orius* is known as voracious predator of many lepidopterans (eggs and young larvae), aphids, spider mites, thrips, and whitefly species globally (*Alvarado, Balta & Alomar, 1997*; *Arnó, Roig & Riudavets, 2008*; *Ceglarska, 1999*; *Riudavets & Castañé, 1998*). In addition to being polyphagous, they have successfully adapted to feed on a wide range of insects and hence found in all of the zoogeographic regions of the world. As a result, several species have become effective predators against a wide range of economic insect pests, used as biological control agents and utilizing in an augmentation strategy globally (*Ballal & Yamada, 2016*). In a previous study, the *Orius sauteri* (Heteroptera: Anthocoridae) was used in the greenhouse to manage pests of pepper and eggplant (*Jiang et al., 2011*; *Kageyama et al., 2010*). Several studies have examined the predatory capacity and behavior of *Orius insidiosus*

(Heteroptera: Anthocoridae), *Orius niger* (Heteroptera: Anthocoridae), *Orius albidipennis* (Heteroptera: Anthocoridae), and *Orius majusculus* (Heteroptera: Anthocoridae) against numerous pest species (*Fritsche & Tamo, 2000*; *Rutledge & O'Neil, 2005*; *Tommasini, Van Lenteren & Burgio, 2004*). Many important pests of vegetable crops are being preyed upon by *Orius laevigatus* (Heteroptera: Anthocoridae) in Europe, and it has been used in many biological control programs (*Van Lenteren & Bueno, 2003*). Recent studies have examined the predatory capacity of *Orius minutus* (Heteroptera: Anthocoridae) and *Orius insidiosus* (Heteroptera: Anthocoridae) on egg of many lepidopterans to determine their predatory capacity (*Brito et al., 2009*; *Sun, Yi & Zheng, 2017*).

The predatory bug *Orius strigicollis* Poppius (Heteroptera: Anthocoridae) which was previously known as *Orius similis* Zheng (Heteroptera: Anthocoridae) (junior synonym of *O . strigicollis*) (*Jung, Yamada & Lee, 2013*; *Rehman et al., 2020*; *Yasunaga, 1997*), also called "minute predatory flower bug", is commonly found in the cultivated fields of China and used as the natural enemy of many destructive pests of agricultural crops (*Zhang et al., 2012*). The nymphal and adult stages of this species are both omnivorous predators, eating the eggs and larvae of lepidopterous insects, such as *Helicoverpa armigera* (Lepidoptera: Noctuidae), *Spodoptera exigua* (Lepidoptera: Noctuidae), *Pectinophora gossypiella* (Lepidoptera: Gelechiidae), and *Anomis flava* (Lepidoptera: Erebidae), as well as pollen (*Aragón-Sánchez et al., 2018*; *Zhang et al., 1994*). Additionally, they are also predators of *Aphis gossypii* (Hemiptera: Aphididae), *Tetranychus cinnabarinus* (Acari: Tetranychidae), and *Frankliniella formosae* (Thysanoptera: Thripidae). It is thought that *O. strigicollis* may be a crucial biological control agent due to their ability to increase in number against prey density, searching efficiency, and their capacity to aggregate in areas with high prey populations (*Hodgson & Aveling, 1988*). Also, they can produce four to eight generations between March and October, depending on the temperature ranges between 20 °C to 37 °C (*Ali et al., 2020*; *Amer, Fu & Niu, 2018*; *Zhang et al., 2012*; *Zhou & Lei, 2002*; *Zhou et al., 2006*). Consequently, the mass rearing of *O. strigicollis* and their release into fields or greenhouses can assist in the control of many important agricultural pests (*Bonte & De Clercq, 2011*).

Insects are very sensitive to their environment, especially temperature, which has a significant impact on their growth, development, survival, and feeding activity. It has also been shown that temperature changes can directly affect predator metabolism, and thus, disrupt their feeding activity (*Angilletta Jr & Angilletta, 2009*; *Gilioli, Baumgärtner & Vacante, 2005*; *May, 1979*; *Parajulee et al., 2006*). An effective predator can thrive and reproduce within agricultural ecosystems, even under adverse conditions (*Cocuzza et al., 1997*). Previous studies have revealed that predators have changed their activity along with the temperature changes, even at different stages of their life (*Jalali, Tirry & De Clercq, 2010*; *Khan et al., 2016*). Similarly, these conditions not only influence the development of insects, but also play an important role in determining the rate at which predators consume their prey. The majority of biological control agents are produced inside controlled laboratory conditions, where they have a high degree of success in surviving, reproducing, and developing (*Bigler, 1994*) but they have less success in controlling prey populations when released in the field. Thus, it is crucial to ascertain the optimal temperature range at
which predators can effectively control pests before their release. Therefore, current study was designed to identify the suitable temperature for releasing of *O. strigicollis* against *P. xylostella* in the field. Until now, a very small number of studies have investigated the use of predators against *P. xylostella* (*Furlong et al., 2004*; *Ma et al., 2005*). In addition, the feeding rate of the *O. strigicollis* and its effect on *P. xylostella* population has not been described in the literature (*Furlong et al., 2004*). Thus, the main objective of this study was to determine the predation potential along with its biological traits, including development time, longevity, oviposition and pre-oviposition periods, and survival rate, when *O. strigicollis* fed on *P. xylostella* eggs at two temperatures (28 °C and 32 °C). Furthermore, the demographic parameters ($r$, $\lambda$, $R_0$, $T$ and $GRR$) were also estimated. This study aimed to contribute information on the potential use of *O. strigicollis* in integrated pest management (IPM) programs against *P. xylostella*.

## MATERIALS & METHODS

### Insects

A few individuals of *O. strigicollis* were collected from the experimental cotton fields of Huazhong Agricultural University (Wuhan, China). A further stack culture was setup following the same method that we used in our previous study (*Rehman et al., 2020*), described by *Zhou et al. (2006)* with modifications. The rearing arenas were made of small plastic containers with lids, each measuring 23 cm × 22 cm × 5 cm. Both young and adults individuals of predatory bugs were supplied with *Aphis fabae* to eat and two to three stems of *Vitex negundo* (Lamiaceae), commonly known as Chinese chaste, were wrapped in wet cotton at the end to facilitate the oviposition. Moreover, the leaves of *V. negundo* provide a microclimate that can be suitable for the development of *Orius* eggs and nymphs, offering optimal humidity and temperature conditions (*Ali et al., 2020*; *Zhou et al., 2006*). All the bugs were reared in an insectary located on the campus with the following environmental conditions; temperature 26 ± 1 °C, relative humidity (RH) 70 ± 5%, and a photoperiod of 16 L: 8 D h at a light intensity of 1,400-1,725 lux.

Eggs of *P. xylostella* were collected from Hubei Key Laboratory of Insect Resources Utilization and Sustainable Pest Management, College of Plant Science and Technology, Huazhong Agricultural University, Wuhan, China. The newly hatched larvae were provided with Chinese cabbage leaves that were sown in small plastic pots and placed inside large screen cages (65 × 65 × 65 cm) until pupate. The pupae then transferred to new cages for adult emergence. After emergence, adult were provided with honey solution (10%). The radish (*Raphanus sativus*: Brassicales: Brassicaceae) were potted in small plastic boxes (6 × 10 × 20 cm) and provided to adults for oviposition. The temperature and relative humidity for *P. xylostella* mass rearing were set at 25 ± 1 °C and 75 ± 5% (RH) respectively with 16 L: 8D light and dark photoperiod.

### Experimental protocols

The mean temperature in the area of the Yangtze River is subtropical (*Zhang et al., 2012*). The results obtained from the study conducted by *Cocuzza et al. (1997)* on *O. albidpennis*, showed that *Orius* spp. can adapt to a range of high temperatures. Hence, based on the

previous studies conducted on *O. strigicollis* in this region (*Ali et al., 2020*; *Amer, Fu & Niu, 2018*; *Zhang et al., 2012*; *Zhou et al., 2006*), two temperatures (*i.e.,* 28 and 32 $\pm$ 1 °C) were selected for our study. The artificial climate controlled chamber (HP250GS, Ruihua Instrument & Equipment Co., Ltd., Wuhan, China) with 70 $\pm$ 5% R.H and 16: 8 L & D photoperiod was used in all experiments.

## Bioassay

The biological characteristics of *O. strigicollis* were determine using the same method we used in the previous study (*Rehman et al., 2020*). In total, 80 fresh eggs of predatory bugs were collected from the insectary, incubated, and after hatching the individuals of *O. strigicollis* ($\leq$ 24 h) were separated in small Petri dishes (diameter: nine cm and depth: two cm) containing filter paper (*i.e.,* experimental unit). A small stem of *Vitex negundo* was provided in each arena to avoid desiccation. Based on the preliminary experiment, 30 eggs of *P. xylostella* were introduced to each experimental unit as food. The number of eggs consumed by predatory bugs in whole or part was counted after every 24 h. Nymphal development and mortality were also recorded. The gender was confirmed immediately after adult emergence. Once the adults had emerged, the male and female gender of *O. strigicollis* were paired for mating. The females of predatory bugs that remained in mating for >1.5 min were considered as mated (*Butler & O'Neil, 2006*). After mating, each pair was transferred into small limpid boxes as described above in the mass rearing of *O. strigicollis*. Each box was supplied with a small tender stem of *Vitex negundo* as oviposition substrate at each temperature (28 °C and 32 °C). The healthy eggs of *P. xylostella* ($n = 60$) were provided to each pair of *O. strigicollis* for feeding every day. The number of eggs consumed by *O. strigicollis* adults was counted and replaced by new eggs every day. To confirm the oviposition, the stem was examined every 24 h under a stereomicroscope (15×). After the first egg was laid by a female, the stem was replaced every day. The total number of eggs laid by a female was counted under a stereomicroscope (15×) until her death. The daily egg consumption and life table traits such as developmental time, pre-oviposition and oviposition, fecundity, longevity, and survival of male and female adults of the predatory bug were recorded at each temperature.

## Statistical analysis

To calculate daily and total egg consumption, the data were analyzed by independent samples $t$-test without transformation ($P < 0.05$). Data were analyzed using SPSS (version 23, SPSS Inc., Chicago, IL, USA).

The data obtained from life table study was analyzed following the same method that was used in our previous study (*Rehman et al., 2020*). This method based on the age-stage, two-sex life table theory using a computer program (Twosex-MSChart). (*Butler & O'Neil, 2006*; *Chi, 1988*). The population and demographic parameters (eggs, nymphs and adult development, oviposition, pre-oviposition, fecundity, and $r$, $\lambda$, $R_0$, *GRR* and $T$) were calculated using the method described by *Chi (2015)*. Equations (1) and (2) was used to

calculate the age-specific survival rate ($l_x$) and fecundity ($m_x$).

$$l_x = \sum_{j=1}^{k} s_{xj} \tag{1}$$

$$m_x = \frac{\sum_{j=1}^{k} s_{xj} f_{xj}}{\sum_{j=1}^{k} s_{xj}}. \tag{2}$$

In the equation, $S_{xj}$ represents age-stage specific survival rate of predatory bug ($x$ = age in days, and $j$ = stage) which indicates the probability of survival of infant to age $x$ and stage $j$. similarly, $f_{xj}$ donates age-stage specific fecundity of adult female (*Chi & Liu, 1985*). To calculate the intrinsic rate of increase ($r$), the Euler–Lotka equation was used, following the system of iterative division with the age index from 0 using Eq. (3) (*Goodman, 1982*).

$$\sum_{x=0}^{\infty} e^{-r(x+1)} l_x m_x = 1. \tag{3}$$

The values of ($R_0$) (the potential of a female to produce a mean number of progenies during her life), was determine using Eq. (4).

$$\sum_{x=0}^{\infty} l_x m_x = R_0. \tag{4}$$

The relation between female and $R_0$ can describe as

$$R_0 = F \frac{N_f}{N}. \tag{5}$$

In the equation $N$ and $N_f$ donate the total number of *O. strigicollis* used in the experiment and the number of adult female respectively (*Chi, 1988*). The gross reproduction rate (*GRR*) and rate of increase ($\lambda$) were calculated using Eqs. (6) and (7).

$$GRR = \sum_{x=0}^{\infty} m_x \tag{6}$$

$$\lambda = er. \tag{7}$$

The mean generation time ($T$) that a population stand in need to rise to $R_0$-fold of its size *i.e.*, $e^{rT} = R_0$ or $\lambda^T = R_0$ at the stable age-stage distribution was determine using the Eq. (8).

$$T = \frac{\ln R_0}{r}. \tag{8}$$

To calculate the stander errors (SEM) of population and demographic parameters, a bootstrap test method (100,000 bootstrap) was used (*Akca et al., 2015*; *Akköprü et al., 2015*; *Tuan et al., 2015*). To compare the means, a pair bootstrap test method built on the confidence of the interval of difference was used (*Chi, 2015*; *Pru et al., 2015*). To get the peaks of survival rate, fecundity, life expectancy and reproductive vales curves, Sigma Plot 12.0 was used.

## RESULTS

### Predation rate

The effect of two temperatures on daily and total egg consumption of *P. xylostella* by *O. strigicollis* is given in Fig. 1. The daily egg consumption of the first and second instars nymph were significantly higher at 28 °C ($F$: 0.024, $t$: 5.071, $df$: 69, $P$: 0.0001) than at 32 °C ($F$: 0.564, $t$: 4.317, $df$: 42, $P$: 0.0001). However, the rest of the nymphal instars showed no significant difference in daily predation at both temperatures. While considering the entire pre-adult stage of life, the total egg consumption of third and fifth nymphal instars were highest ($F$: 33.262, $t$: −2.126, $df$: 53.630, $P$: 0.038; $F$: 0.039, $t$: 2.880, $df$: 30, $P$: 0.007), with 36.3 eggs per individual at 32 °C compared to 25.8 eggs at 28 °C. Whereas, the consumption of the total eggs by an adult *O. strigicollis* was statistically higher (68 eggs; $F$: 1.633, $t$: 2.872, $df$: 62, $P$: 0.006) at 28 °C. The maximum daily predation (15 eggs; $F$: 6.529, $t$: −4.523, $df$: 62, $P$: 0.000) at the adult stage was recorded at 32 °C.

### Nymphal development

The effect of two temperatures on growth and development of *O. strigicollis*, when fed on eggs of *P. xylostella*, is shown in Table 1. The obtained results showed that the mean developmental time for eggs of the predatory bug was significantly shorter at 32 °C (2.40 days) compared to 28 °C (2.83 days). The developmental time for 1st, 2nd, and 5th instar nymphs of *O. strigicollis* was significantly increased at 28 °C. Similarly, the female adult longevity was statistically different and higher (17.33 ± 2.30 days) at 28 °C. However, no significant difference was found in male adult longevity (9.12 ± 1.59 and 7.29 ± 1.90 days) at 28 and 32 °C, respectively. The highest survival rate (42%) of individuals from the eggs to adult stage of life was recorded at 28 °C than at 32 °C (28%).

### Fecundity and oviposition

A significant difference was also noticed in the adult pre-oviposition period (APOP), total pre-oviposition period (TPOP), oviposition period, and fecundity of female adult *O. strigicollis* (Table 2). The highest pre-oviposition (5.0 days) and oviposition period (9.38 days) was recorded at the 28 °C. Similarly, the total fecundity per female adult of *O. strigicollis* was highest (42 eggs/female) at 28 °C, whereas, it was 17.3 eggs/female at 32 °C. Moreover, the total longevity of adult females was longer (33.44 ± 1.32) at 28 °C. However, no significant difference was observed in the male adult total longevity at both temperatures.

### Life table and population parameters

The highest intrinsic rate ($r$: 0.087 d$^{-1}$) was observed at 28 °C (Table 3). Thus, our results showed that $r$ decreased significantly with the increase in temperature. Whereas, the rate of increase significantly decreased to λ: 1.09 d$^{-1}$ at 28 °C compared to at 32 °C (1.28 d$^{-1}$). The net reproductive rate was $R_0$: 9.45 and 1.28 offspring/individual at 28 °C and 32 °C respectively. The mean generation time ($T$) and gross reproduction rate ($GRR$) significantly increased from 25.71 days, 38.70/offspring and 19.15 days, 8.12/offspring with the increase in temperature from 28 °C to 32 °C, respectively.

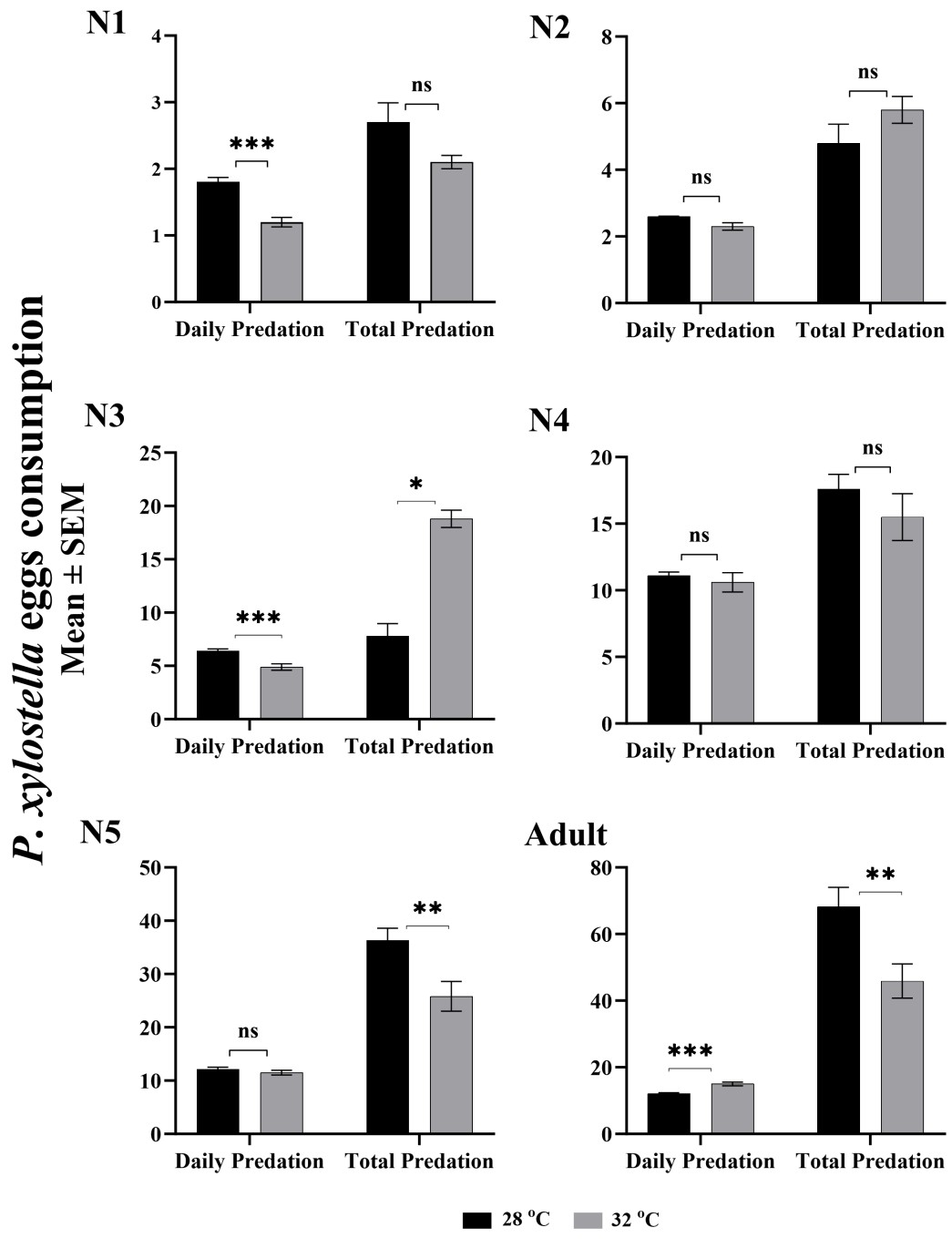

**Figure 1** **Stage-specific daily and total predation of *O. strigicollis* fed on eggs of *Plutella xylostella* at two temperatures.** The value of each bar represent the mean eggs consumption, and the error bars indicates the S.E.M, asterisk indicates a significant difference following the independent samples $t$-test (These asterisks (*,**,**) represents the significance level according to the independent t-test which is equal to $P < 0.05$.).

The age-stage specific survival rate ($S_{xj}$, $x$ = age in days while $j$ = stage which describes the probability of survival of a neonate to age $x$ and stage $j$) under two different temperatures

**Table 1** **Influence of two temperatures on growth and development of _O. strigicollis_ fed on eggs of _P. xylostella_.** The developmental time for N1- N5 was significantly higher at 28 °C. Similarly, female adult longevity was also highest at 28 °C.

| Stage duration (days) | Temperatures | | | |
|---|---|---|---|---|
| | 28 °C | | 32 °C | |
| | _n_ | Mean ± S.E. | _n_ | Mean ± S.E. |
| Egg | 40 | 2.83 ± 0.06 a | 40 | 2.40 ± 0.08 b |
| N1 | 29 | 2.62 ± 0.09 a | 25 | 2.00 ± 0.00 b |
| N2 | 23 | 3.13 ± 0.1 a | 19 | 2.63 ± 0.14 b |
| N3 | 21 | 2.24 ± 0.12 a | 16 | 2.38 ± 0.12 a |
| N4 | 20 | 1.65 ± 0.11 a | 13 | 1.69 ± 0.17 a |
| N5 | 17 | 3.18 ± 0.18 a | 10 | 2.30 ± 0.21 b |
| N1-N5 | 17 | 13.18 ± 0.26 a | 10 | 11.10 ± 0.23 b |
| Female adult longevity | 9 | 17.33 ± 2.30 a | 3 | 8.33 ± 2.73 b |
| Male adult longevity | 8 | 9.12 ± 1.59 a | 7 | 7.29 ± 1.90 a |

Notes.
SEs were estimated using 100,000 bootstrap. Means marked with multiple letters in the same row symbolize the significant difference using a pair bootstrap test. $P < 0.05$.

**Table 2** **Influence of two temperatures on biological traits of O. strigicollis fed on eggs of _P. xylostella_.**

| Parameters | Temperatures | |
|---|---|---|
| | 28 °C | 32 °C |
| | Mean ±S.E. | Mean ±S.E. |
| APOP (d) | 5.0 ± 0.50 a | 2.5 ± 0.58 b |
| TPOP (d) | 21 ± 0.71 a | 15.50 ± 1.50 b |
| Oviposition (d) | 9.38 ± 0.71 a | 6.0 ± 0.00 b |
| Fecundity (total eggs/female) | 42 ± 6.26 a | 17.3 ± 8.54 b |
| Mean longevity of adult Female | 33.44 ± 1.32 a | 21 ± 3.00 b |
| Mean longevity of adult Male | 25.38 ± 1.32 a | 20.86 ± 1.92 a |
| Sex ratio (F:M) | 9:8 | 3:7 |

Notes.
APOP, Adult pre-oviposition period of female adult;
TPOP, Adult pre-oviposition period of female counted from the birth.
SEs were estimated using 100,000 bootstrap. Means marked with multiple letters in the same row symbolize the significant difference using a pair bootstrap test $P < 005$.

when fed on eggs of _P. xylostella_ is given in Fig. 2. The results showed variations in the developmental rate and overlapping occurred at the start and end of each stage at both temperatures. The probability of survival of a newly laid female egg to the adult stage of life was 22.5% and 7.5% at 28 °C and 32 °C temperatures respectively. Similarly, for male eggs, the probability of surviving to adulthood was 20% and 17.5% at 28 °C and 32 °C, respectively.

The curve of age-specific survival rate ($l_x$), the age-specific total fecundity of the whole population ($m_x$), the age-stage based female fecundity ($f_x$), and age-specific maternity ($l_x m_x$: formed on the basis of $l_x$ & $m_x$) at two temperatures when _Anthocorid_ spp. fed on

**Table 3 Influence of two temperatures on demographic parameters of O. strigicollis fed on eggs of P. xylostella.** The demographic parameters showed significant increase at 28 °C.

| Parameters | Temperatures | |
| --- | --- | --- |
| | 28 °C | 32 °C |
| | Mean ±S.E. | Mean ±S.E. |
| Intrinsic rate of increase ($r$) ($d^{-1}$) | $0.087 \pm 0.015$ a | $0.013 \pm 0.028$ b |
| Rate of increase ($\lambda$) ($d^{-1}$) | $1.091 \pm 0.016$ a | $1.28 \pm 0.029$ b |
| Net reproductive rate ($R_0$) (offspring/individual) | $9.45 \pm 3.081$ a | $1.28 \pm 0.784$ b |
| Generation time ($T$) (days) | $25.71 \pm 0.769$ a | $19.15 \pm 1.213$ b |
| Gross reproduction rate ($GRR$) (offspring) | $38.70 \pm 6.560$ a | $8.12 \pm 4.124$ b |
| Intrinsic rate of increase ($r$) ($d^{-1}$) | $0.087 \pm 0.015$ a | $0.013 \pm 0.028$ b |

**Notes.**

SEs were estimated using 100,000 bootstraps. Means marked with different letters in the same row symbolize the significant difference using a pair bootstrap test. $P < 005$.

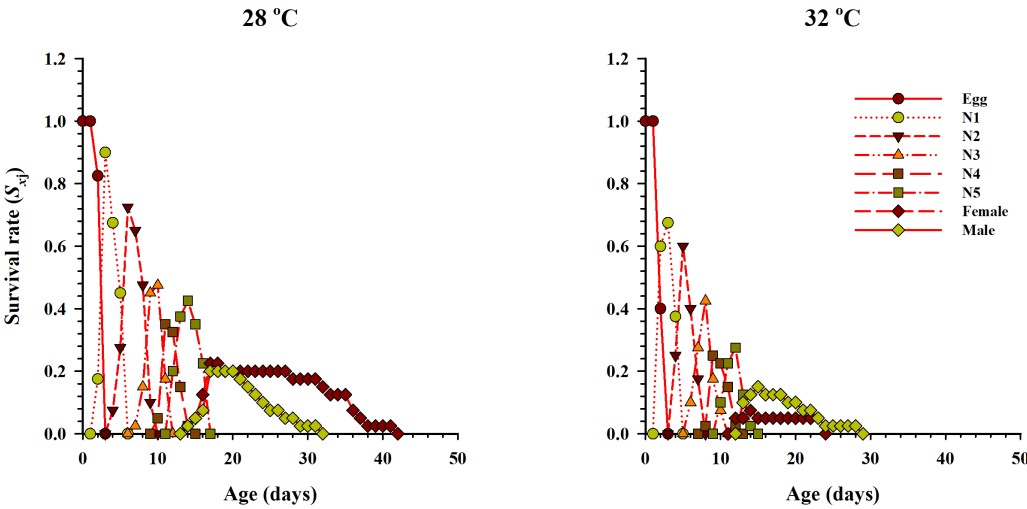

**Figure 2 Age-stage-specific survival rate ($S_{xj}$) of O. strigicollis at two temperatures fed on P. xylostella eggs.** The probability of survival of a newly laid female egg to the adult stage of life was highest at 28 °C.

eggs of *P. xylostella* is shown in Fig. 3. In the life table study, ($l_x$) and ($m_x$) are considered essential parameters. Age-specific survival ($l_x$) decreased with age and showed an inverse relation with low and high regimes. However, $f_x$, $m_x$ and $l_x m_x$ first increased and then decreased as age increased at both temperatures. The highest value of $m_x$ and $f_x$ (3.62 eggs and 5.1 eggs) was calculated at 28 °C.

The curve of age-stage specific life expectancy ($e_{xj}$) for *O. strigicollis*, when fed on eggs of *P. xylostella*, affected by different temperatures is shown in Fig. 4. Results showed an inverse relation for life expectancy with temperature as it decreased from 16.96 days to 10.04 days at 28 °C and 32 °C respectively. The age-stage reproductive value ($v_{xj}$ *i.e.,* the forecasting scale for future population individuals of *O. strigicollis* at age $x$ and stage $j$) under two temperatures is given in Fig. 5. The peak of age-stage specific curve ($V_{xj}$) for newly laid eggs was 1.51 eggs and 1.53 eggs at 28 °C and 32 °C, respectively. The curve of reproductive
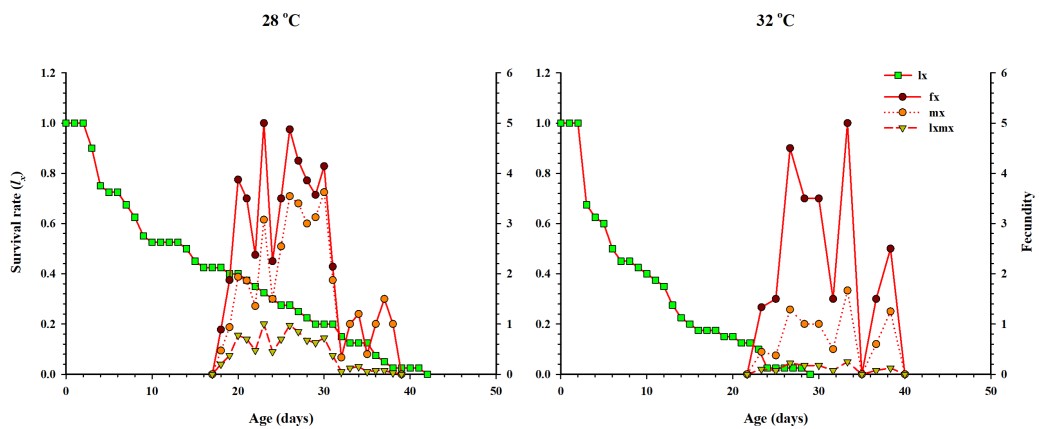

**Figure 3** **Age-specific survival rate ($l_x$), female age-specific fecundity ($f_x$), age-specific fecundity of the total population ($m_x$), and age-specific maternity ($l_x m_x$) of *O. strigicollis*.** $l_x$, Age-specific survival rate; $f_x$: female age-specific fecundity; $m_x$: age-specific fecundity of the total population; $l_x m_x$: and age-specific maternity.

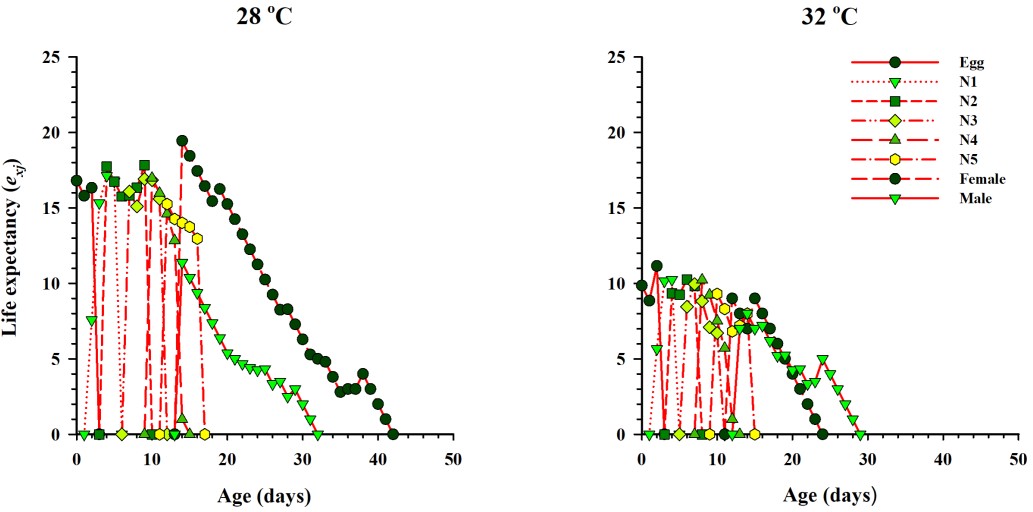

**Figure 4** **Age-specific life expectancy ($e_{xj}$) of *O. strigicollis* on *P. xylostella* eggs at two temperatures.** Results showed an inverse relation for life expectancy with temperature as it decreased from 28 °C to 32 °C.

values significantly increased as the age and stage increased at each temperature. The highest peak for an adult female of *O. strigicollis* was obtained on the 20th and 15th day at 28 °C and 32 °C respectively.

## DISCUSSION

The present study concluded that *O. strigicollis* could develop and survive at both temperatures when fed on *P. xylostella* eggs. Except 1st, 3rd and 5th nymphal instars, no significant effect of temperature was recorded on predation. However, *O. strigicollis*

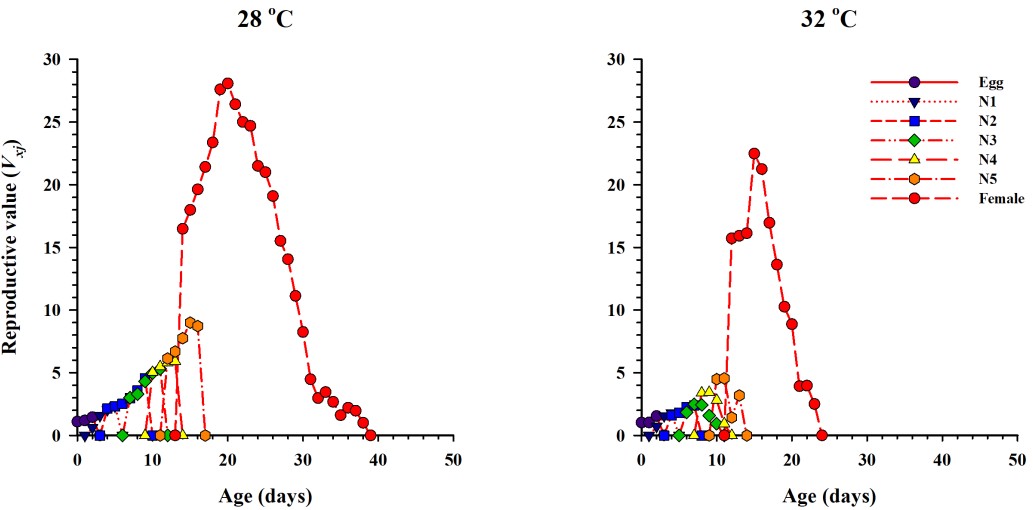

**Figure 5  Age-stage specific reproductive values ($V_{x}j$) of *O. strigicollis* on *P. xylostella* eggs at two temperatures.** The curve of reproductive values significantly increased as the age and stage increased at each temperature.

adults consumed significantly more eggs at 32 °C. It is possible that the difference in consumption was caused by the large body size and the high nutritional requirements of adults stage (*Aragón-Sánchez et al., 2018*). The results of previous studies were in agreement with our results when *O. similis* was tested at different temperatures (20, 25, 30 and 35 °C), and the *Aphis gossypii* consumption was maximum at 35 °C (*Zhou et al., 2006*). Comparing our results with *Brito et al. (2009)*, the *O. insidiosus* predation capacity on *P. xylostella* eggs was lower than our results. While, the mean daily consumption of *O. minutus* and *O. laevigatus* on eggs of *P. xylostella* and *S. exigua* was higher than the number obtained in the present study (*Aragón-Sánchez et al., 2018*; *Sun, Yi & Zheng, 2017*).

The results obtained from the age-stage two-sex life table analysis showed that the predatory bug *O. strigicollis* successfully survived and developed at each tested temperature when fed on eggs of *P. xylostella*. However, the eggs and nymphal development of Anthocorid spp. were significantly different and shorter when the temperature was 32 °C. In recent studies, *O. similis* eggs developed relatively longer (3.9, 3.4 and 3.2 days) when fed on *Aphis gossypii* and *Myzus persicae* at 28, 30 and 31 °C, respectively (*Ahmadi, Sengonca & Blaeser, 2007*; *Zhou et al., 2006*). Whereas, the development of *O. albidipennis* eggs at 25, 30 and 35 °C, when fed on *F. occidentalis* and *M. sjostedti*, was 3.8, 2.9 and 2.7 days, respectively (*Cocuzza et al., 1997*; *Gitonga et al., 2002*). The above-cited studies are supporting our results and indicate that the egg development of the predatory bug of the genus *Orius* decreased with an increased temperature. Moreover, *O. strigicollis* pre-adults also showed sensitivity to different temperatures. The nymphal development of predatory bug was short at 32 °C when fed on eggs of *P. xylostella*. The same interaction was documented in numerous studies where the fluctuations in temperatures influenced nymphal development. Relating our results with *Zhou et al. (2006)*, the nymphal development of *O. similis* was short and similar at 32 °C when tested against five different temperatures. Similarly,

when fed on different prey species, the mean developmental duration of *O. laevigatus*, *O. similis*, *O. majusculus* and *O. albidipennis* significantly decreased with increased in temperature (*Ahmadi, Sengonca & Blaeser, 2007*; *Cocuzza et al., 1997*; *Martínez-García et al., 2018*). During the exposure to 28 °C and 32 °C temperatures, *O. strigicollis'* adult longevity was greatly influenced. A significant increase in female adult longevity was observed at 28 °C, while no significant differences were observed in male adult longevity at either temperature. An increase in the temperature significantly affected the longevity of *O. sauteri* and *O. strigicollis* adults when they were fed different prey species (*Nagai & Yano, 1999*; *Ohta, 2001*). Thus, these variations in the developmental duration of eggs, pre-adults and adults of *O. strigicollis* may directly be associated with the change in the type of prey they consumed at different temperatures (*Jaleel, Lu & He, 2018*; *Sengonca, Ahmadi & Blaeser, 2008*). The results further showed that pre-adult survival from eggs to adulthood decreased when the temperature increased. The highest levels of survival occur at 28 °C. The nymphal survival of *O. strigicollis* decreased from 60% to 37.3% at temperatures between 15 to 30 °C (*Ohta, 2001*). Similarly, *O. albidipennis* nymphal survival was highest at 25 °C when tested against three different temperatures (*Gitonga et al., 2002*).

The adult pre-oviposition period (APOP) showed an inverse relation at both temperatures and significantly decreased at 32 °C. A similar inclination was reported in previous studies when the length of the pre-oviposition period of *O. strigicollis* decreased from 15.6 to 4.7 days at 17 °C and 29 °C, respectively (*Kakimoto et al., 2005*). Similarly, the recorded pre-oviposition period for *O. similis* was shorter (5.5 days) at 31 °C, when fed on *T. cinnabarinus* at three constant temperatures (*Zhang et al., 2012*). In a study documented by *Ahmadi, Sengonca & Blaeser (2007)*, the oviposition period of female adult *O. similis* was significantly highest at 30 °C compared to at 18 °C when fed on two different species of aphid. Furthermore, *O. similis* oviposition period decreased from 21.1 to 18.8 days when fed on *T. cinnabarinus* at 28 °C and 31 °C (*Zhang et al., 2012*). Our results are in agreement with the above-cited studies and show that the oviposition period of *O. strigicollis* was significantly high at 28 °C and gradually shortened with an increase in temperature. In the present study, the mean total fecundity of female adult *O. strigicollis* was affected by *P. xylostella* eggs as prey at both temperatures (28 and 32 °C). In previous studies, many authors confirmed similar changes in fecundity with different prey species and temperatures (*Fritsche & Tamo, 2000*; *Sengonca, Ahmadi & Blaeser, 2008*; *Tommasini, Van Lenteren & Burgio, 2004*). Present study showed that *O. strigicollis* mean fecundity decreased at 32 °C. *Zhang et al. (2012)* reported similar results, when the total fecundity of *O. similis* decreased from 40.3 to 34.3 eggs at 28 °C and 31 °C supplied with spider mites.

Life table studies enable us to understand the ecology of an organism. They provide some crucial tools for assessing the effect of different temperatures on biological traits and population parameters. Our results showed an inverse relationship in age-stage specific fecundity ($m_x$), survival rate ($S_{xj}$), life expectancy ($e_{xj}$) and reproductive value ($v_{xj}$). As the age and stage of *O. strigicollis* increase, all the parameters dramatically decreased at both temperatures. The effect of two different temperatures on *O. strigicollis*, when fed on *P. xylostella* eggs, was further substantiated by the intrinsic rate of increase ($r$). The intrinsic rate of increase ($r$) and net reproductive rate ($R_0$) are considered crucial population

parameters to determine the fitness traits of an organism such as growth, development, survival, and its higher values ($r > 0$ and $R_0 > 1$) confirm the suitability of prey with its predator with increased in the mean population (*Chen et al., 2017*; *Southwood & Henderson, 2009*). Moreover, using the intrinsic rate of increase, we can get the effects of fertility and mortality factors into a single value. Consequently, a single comparison can be applied between populations instead of comparing the several biological characteristics (*Havelka & Zemek, 1999*). Our results were according to the above-mentioned theories and $r$ and $R_0$ were significantly different and highest at 28 °C. In contrast, the calculated values of $r$ and $R_0$ for *O. niger* were comparatively higher, when fed on eggs of *Ephestia kuehniella* at 29 and 32 °C (*Baniameri, Soleiman-Nejadian & Mohaghegh, 2005*). Similar results were reported when *O. similis* and *O. sauteri* fed on *T. cinnabarinus* and *T. palmi* as prey under different temperatures (*Nagai & Yano, 1999*; *Zhang et al., 2012*). The gross reproduction rate (*GRR*) is considered an indicator of population growth and is associated with the number of eggs laid and hatched and adult eclosion (*Cocuzza et al., 1997*; *Huang & Chi, 2013*). In our results, the *GRR* and *T* were significantly different and higher at 28 °C. A similar reduction in *T* and *GRR* was observed when *O. similis, O. laevigatus* and *O. albidipennis* were tested at different temperatures (*Cocuzza et al., 1997*; *Zhang et al., 2012*). However, the documented values for *T* and *GRR* by several authors were significantly changed and increased with temperatures when different species of *Orius* were tested at several constant temperatures (*Hamdan, 2015*; *Hamdan, 2012*; *Kakimoto et al., 2005*; *Tommasini, Van Lenteren & Burgio, 2004*).

## CONCLUSIONS

In summary, our study concluded that *O. strigicollis* can grow, developed, reproduce and prey on *P. xylostella* eggs at both temperatures. However, the predatory bug showed relatively good survival, maximum predation and female fecundity at 28 °C. Thus, we can suggest that the inoculative release of *O. strigicollis* at 28 °C against *P. xylostella* could suppress the *P. xylostella* population more progressively. Laboratory experiments should help integrated pest management (IPM) strategies. However, field experiments are needed to confirm the hypotheses constructed from the present study.

### Funding

This research was funded by the National Natural Science Foundation of China, Grant No. 31872023 and the Deanship of Scientific Research at King Khalid University Saudi Arabia, Grant No. RGP2/328/45. The funders had no role in study design, data collection and analysis, decision to publish, or preparation of the manuscript.

### Grant Disclosures

The following grant information was disclosed by the authors:
National Natural Science Foundation of China: 31872023.
Deanship of Scientific Research at King Khalid University Saudi Arabia: RGP2/328/45.

## Competing Interests

The authors declare there are no competing interests.

## Author Contributions

- Shakeel Ur Rehman performed the experiments, analyzed the data, prepared figures and/or tables, and approved the final draft.
- Xingfu Jiang analyzed the data, authored or reviewed drafts of the article, and approved the final draft.
- Mahnoor Saleem analyzed the data, prepared figures and/or tables, and approved the final draft.
- Xingmiao Zhou conceived and designed the experiments, performed the experiments, authored or reviewed drafts of the article, and approved the final draft.
- Bangqing Chen analyzed the data, authored or reviewed drafts of the article, and approved the final draft.
- Khalid Ali Khan analyzed the data, prepared figures and/or tables, and approved the final draft.
- Ibrahim Osman Ibrahim analyzed the data, prepared figures and/or tables, and approved the final draft.
- Hamed A. Ghramh analyzed the data, authored or reviewed drafts of the article, and approved the final draft.

## Data Availability

The raw data are available in the Supplemental Files.

## Supplemental Information

Supplemental information for this article can be found online at http://dx.doi.org/10.7717/peerj.18044#supplemental-information.

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
