# Peer review of "Demography and predatory potential of Orius strigicollis on eggs of Plutella xylostella at two temperatures"

_PeerJ, doi:10.7717/peerj.18044_

## Round 0.1 · original submission · Major Revisions

This manuscript is a worthy contribution to the field, however reviewers have numerous concerns which must be addressed before the manuscript can be accepted. Please pay particular attention to the issues raised by Reviewer 1 and give detailed responses to all reviewers' concerns.

Reviewer 1 ·

Basic reporting

The study concerns the biological characteristics and predation capacity of the Anthocoridae species Orius strigicollis on the eggs of the pest Lepidoptera species Plutella xylostella at two different temperatures. However, since the term Anthocorid spp. is frequently mentioned in the article, this causes confusion. Firstly, this needs to be clarified.
Additionally, the Latin names of many species should be fully written out the first time they are mentioned, without abbreviation. If possible, English names should be added, along with the order and family, including the authors' names (Lines 79-81, Lines 90-92).
Line 110-11: Therefore, current study was designed to identify the optimal temperature for releasing Orius. Could you explain this sentence? Firstly, there are many ecological factors that affect the feeding of a predator on any species under field conditions. Working at two different temperatures should not be the sole criterion for predator releases. Temperatures in natural conditions vary widely. It may be difficult to adjust these temperatures for predator releases in field conditions, as the development of the predator insect primarily depends on prey density, which can vary according to temperature conditions.

Experimental design

Expermental design is well-planned, Howewer there are a few points that should be explaned, given below.
Line 128: Vitex negundo, please give brief information about this plant species i.e. why did you select this plant species as an oviposition substrate for rearing of the minute predatory bug, adding English name, family etc.
Line 143: Please describe which previous studies?

Validity of the findings

Results have impact and novelity, providing some good results but there is a serious mistake to compare means of predatory capacities of nymplal and adult stages, refeering the Figure 1. I have explained them in a detail below.
Line 227-235: You should add results of statistical analysis results such as F, df, P values in all main text.
In Result section, there is no mention for Figure 1 i.e. stage-specific daily and total predation of predatory bugs’s nymphs and adults fed on egged of prey insect species at two different temperature.
Additionally, I would like to point out that the analysis method used in preparing Figure 1 is incorrect. The Tukey multiple comparison method was used to compare the daily and total number of eggs consumed by nymphs and adults at two different temperatures. However, for each assessment (daily and total predation capacities), the mean values at two different temperatures should be compared. Therefore, an appropriate analysis method for paired comparison (such as the t-test) should be selected and the data should be reanalyzed.
Lines 278-288 This paragraph includes general information about this issue, please remove it to Introduction section.
Lines, 297-298: These sentences are not included in discussion because, this is different prey species, thrips (Thysanoptera) which they insert eggs into plant tissues.
Line 374: Anthocorid specie, correct Anthocoridae species.

Additional comments

The authors often used results obtained with different prey species in the interpretation of the results and the discussions with previously published studies. However, there is a study published in the same journal that investigated the same biological characteristics of the same predator species at three different temperatures using pink bollworm as the prey. Comments related to this study were not seen in the discussion section.
The reference list has not been prepared according to the required formatting rules. For example, in some references, journal names are written in lowercase, while in others they are written in uppercase. These inconsistencies need to be corrected.

Reviewer 2 ·

Basic reporting

Overall, the article is quite good. I have made notes and raised questions in the attached DOCX file, for which I would like responses. In the tables, I only adjusted the formatting and suggested some modifications to Figure 1.

Experimental design

is ok

Validity of the findings

they are valid

Annotated reviews are not available for download in order to protect the identity of reviewers who chose to remain anonymous.

Reviewer 3 ·

Basic reporting

I would like to express my sincere gratitude for providing me with the opportunity to review the manuscript submitted to PeerJ. After a careful examination, I have found this study to hold significant merit, particularly in its exploration of an innovative approach to mitigating the Demography and predatory potential of Orius strigicollis on eggs of Plutella xylostella at two temperatures. The author conducted an interesting study to investigate the impact of different temperatures and Plutella xylostella eggs on Orius strigicollis predation, growth and development, fecundity, and other population parameters using age stage Two-Sex Lifetable. Their findings reveal that the predation rate on P. xylostella eggs, the egg-laying capacity and oviposition period of adult female O. strigicollis as well as other population parameters also explain that O. strigicollis would survive until adulthood at 28 oC temperature.

Experimental design

Reasonable experimental design.

Validity of the findings

This research underscores the potential of O. strigicollis against P. xylostella in the biological control program with the suitable temperature to release into the field according.

Additional comments

To further enhance the quality and impact of this work, I would like to offer some suggestions:

Line 88: Contradiction…… Larvae or nymphs stages of O. strigicollis…clear it
Line 91: change “…….In addition to this ……x”…… Additionally
Line 99: change “…….Insects are very sensitive to environmental conditions ……x”…… sensitive to their environment”
Line 101: change “……Predators metabolisms…..x”…..Predator metabolism”.
Line 105: change “………..play a role ........x”….. Play an important role”
Line 105: change “………..release in the open ........x”….. released in the field”
Line 121: The introduction to your methodology section should begin by restating the research problem and underlying assumptions underpinning your study rather than just include subsections. Please, write an introduction, and, at same time, authors can describe the methodology subsections.
Line 132: describe environmental conditions for P. xylostella mass rearing.
Line 138, 139: “correct the sentence structure”
Line 151: “correct the sentence”
Line 160: ‘The females of Anthocorid spp. that remained in mating for >1.5 min were considered as mated’……. “Add reference”
Line 220: change “………..statistical difference ........x”….. significant difference”
Line 278-279: is not necessary (text best in introduction)
Line 307: Add “…oC… with 28, 30”
Line 308: “similar to line 307”
Line 319: “add… oC… with 28”
Line 333: “add… oC… with 17”
Line 339: “add… oC… with 28”
Line 346: “similar to line 339”
Line 451: italic scientific name
Line 470: italic scientific name
Line 473,474: italic scientific name
Line 467-756: The scientific names are not in italic in all the references.

Annotated reviews are not available for download in order to protect the identity of reviewers who chose to remain anonymous.

---

## Round 0.2 · Minor Revisions

Please address all of Reviewer 1's remaining concerns.

Reviewer 1 ·

Basic reporting

Clear and unambiguous, professional English used throughout
Professional article structure, figures, tables. Raw data shared

Experimental design

Research question well defined, relevant & meaningful. It is stated how research fills an identified knowledge gap

Validity of the findings

All underlying data have been provided; they are robust, statistically sound, & controlled.

Additional comments

Dear Author (s); You have already done some of the corrections addressed by me, now, I have showed, considering my criticisms, especially regarding the evaluation of the data. However, when I re-reviewed the manuscript, I found that some corrections have not been done or overlooked. I have stated these below.
Line 31: The species name of Orius strigallis should be written clearly, and the author, order and family names should be stated where they are first mentioned. These fixes were shown in the previous review. Likewise, in Line 34, Lines 87-91, and Lines 99-100, the authors, orders, and families of the insects must be stated.
In my previous manuscript review; I was asked for information about Vitex negundo, for the oviposition substrate used, in the rearing of the predatory insect, and why this plant was chosen. The author added this in the answer section, but this information is still missing in the materials and methods section. Please add it.
In previous assessment (Lines 110-111) I had indicated that current study was designed to identify the optimal temperature for releasing Orius. Could you explain this sentence? Firstly, there are many ecological factors that affect the feeding of a predator on any species under field conditions. Working at two different temperatures should not be the sole criterion for predator releases. Temperatures in natural conditions vary widely. It may be difficult to adjust these temperatures for predator releases in field conditions, as the development of the predator insect primarily depends on prey density, which can vary according to temperature conditions. Authors have explained this issue, adding references in response to reviewers, but all or some important references have not been included in this section and also reference list, please provide them.

Reviewer 3 ·

Basic reporting

The MS has been fully revised according to the requirements and has reached the acceptance level

Experimental design

good

Validity of the findings

medium

---

## Round 0.3 · accepted · Accept

Thank you for addressing the reviewers' concerns.